# Contribution of Epstein–Barr Virus Lytic Proteins to Cancer Hallmarks and Implications from Other Oncoviruses

**DOI:** 10.3390/cancers15072120

**Published:** 2023-04-02

**Authors:** Mike Dorothea, Jia Xie, Stephanie Pei Tung Yiu, Alan Kwok Shing Chiang

**Affiliations:** 1Department of Paediatrics and Adolescent Medicine, School of Clinical Medicine, The University of Hong Kong, Hong Kong, China; mdor@connect.hku.hk (M.D.); thnxplus@connect.hku.hk (J.X.); 2Division of Infectious Diseases, Department of Medicine, Brigham and Women’s Hospital, 181 Longwood Avenue, Boston, MA 02115, USA; stephanie.pty@gmail.com; 3Harvard Graduate Program in Virology, Boston, MA 02115, USA

**Keywords:** Epstein–Barr virus (EBV), lytic proteins, herpesvirus, oncovirus, cancer hallmarks

## Abstract

**Simple Summary:**

Epstein–Barr virus (EBV) is a gamma-herpesvirus associated with a broad variety of cancers. It employs two modes of viral replication, i.e., latent and lytic cycles in which both can contribute to oncogenesis. Increasing evidence supports the contribution of the lytic cycle in cancer development while only a limited number of lytic proteins have been characterized in detail. This review aims to summarize the currently known tumorigenic properties of the lytic proteins based on their contribution to the hallmarks of cancer and to postulate the roles of some partially characterized or uncharacterized lytic proteins based on their homologs in other herpesviruses and oncoviruses.

**Abstract:**

Epstein–Barr virus (EBV) is a prevalent human gamma-herpesvirus that infects the majority of the adult population worldwide and is associated with several lymphoid and epithelial malignancies. EBV displays a biphasic life cycle, namely, latent and lytic replication cycles, expressing a diversity of viral proteins. Among the EBV proteins being expressed during both latent and lytic cycles, the oncogenic roles of EBV lytic proteins are largely uncharacterized. In this review, the established contributions of EBV lytic proteins in tumorigenesis are summarized according to the cancer hallmarks displayed. We further postulate the oncogenic properties of several EBV lytic proteins by comparing the evolutionary conserved oncogenic mechanisms in other herpesviruses and oncoviruses.

## 1. Introduction

Epstein–Barr virus (EBV) is a prevalent human gamma-herpesvirus that infects >95% of the world’s population. It was first discovered in cells from patients with Burkitt Lymphoma (BL) in 1964 denoting that EBV is an infectious agent involved in the etiology of BL [1]. This finding was a milestone in the field of virology and cancer as EBV was the first recognized human oncogenic virus in history. EBV usually infects the majority of individuals quietly. However, EBV can lead to the development of several B-cell, T-/natural killer cell and epithelial cell malignancies [2]. EBV is transmitted orally infecting epithelial cells in the oral cavity followed by B-cells in the pharyngeal lymphoid tissues [3]. Similar to other herpesviruses, EBV has two alternative modes of viral replication, namely, latent and lytic replication [4]. Latent replication of EBV is normally found in the lymphatic system. The virus can occasionally reactivate into the lytic phase and enter into an active replication cycle that generates infectious viral particles to facilitate transmission. One of the characteristics of EBV is its unique presentation of different latency programs in different tumor types. At least three types of latency programs are recognized in different EBV-associated tumors according to the expression pattern of the EBV genes [5]. These patterns are designated as latency I [Epstein-Barr nuclear antigen (EBNA)-1 and non-coding RNAs such as BamHI A rightward transcripts (BARTs), BART and BHRF1 micro-RNAs (miRNAs) and EBV-encoded small RNAs (EBERs)], latency II [EBNA-1, latent membrane protein (LMP)-1, LMP-2 and non-coding RNAs] and latency III [EBNAs-1, -2, -3A, -3B, -3C and -LP, LMP-1, LMP-2 and non-coding RNAs]. These distinctive expression patterns confer survival advantages for EBV since EBV-latently transformed cells express fewer EBV gene products at their cell surface allowing them to evade cytotoxic immune responses. In addition, various EBV-latent gene products can hijack multiple cellular pathways to provide proliferation and survival advantages to the infected cells [6,7,8,9,10].

The EBV lytic cycle is described as a productive replication cycle during which a cascade of more than 80 viral proteins is expressed. Unlike latent replication where EBV persists as multi-copy episomes, the EBV genome is amplified up to 1000-fold with the production of infectious virions during lytic replication [11,12,13]. While EBV latent proteins have been demonstrated to drive cellular transformation and tumorigenesis [6], increasing evidence suggests that EBV lytic proteins are equally important in contributing to oncogenesis [14,15,16,17]. The proteins expressed during lytic replication are classified into immediate early (IE), early (E) and late (L) lytic proteins based on their roles in virus replication [18,19,20,21]. Immediate early lytic proteins, which include BZLF1 (Zta) and BRLF1 (Rta), are transcription factors immediately expressed during the lytic replication cycle and transactivate the cascade of other viral replication mechanisms. On the other hand, early and late lytic proteins support the replication of viral DNA and the assembly and egress of infectious viral particles, respectively. In some circumstances, the lytic replication cycle may be aborted as indicated by the absence of late lytic proteins implying that no progeny virus is released [22]. A humanized mouse model with enhanced Zta expression had detectable early but not late lytic proteins indicating that the abortive lytic cycle can be observed in vivo [23]. This mutant EBV mouse model develops a more aggressive lymphoma phenotype with partial resistance to T-cell killing [23]. Despite the absence of production of viral progenies, the abortive lytic cycle has also been proposed to be the initial mode of EBV infection providing a proliferative advantage to B-cells and driving tumorigenesis in both B- and epithelial cells [22,24,25,26,27,28,29,30]. Numerous EBV lytic proteins are expressed and can impact immunomodulation and tumor microenvironment [23,27,29,30,31].

Due to the extensive array of EBV lytic proteins being transiently expressed during the lytic replication cycle, the roles of EBV lytic proteins in oncogenesis have largely been uncharacterized. This review will summarize the known oncogenic properties of EBV lytic proteins based on their contribution to the hallmarks of cancer and postulate the oncogenic potential of some lytic proteins based on their functional homologs in other herpesviruses or oncoviruses.

## 2. Overview of Cancer Hallmarks

The concept of ‘hallmarks of cancer’ was first introduced by Hanahan and Weinberg in 2000 and represented the cellular traits through which different cancers acquire the capabilities to survive, proliferate and disseminate. The six cancer hallmarks first described were evading apoptosis, self-sufficiency in growth signals, insensitivity to anti-growth signals, sustained angiogenesis, limitless replicative potential and tissue invasion and metastasis [32]. Two emerging hallmarks and two enabling characteristics, which included deregulating cellular energetics, avoiding immune destruction, genome instability and mutation and tumor-promoting inflammation, were appended to the previous conceptual framework in 2011 [33]. These enabling characteristics and emerging hallmarks support tumor progression [33]. Although the concept provided a seminal understanding on oncogenesis, arguments on the inclusion or exclusion criteria of some of the hallmarks emerged. A new definition was proposed by Fouad et al. in 2017 to incorporate evolutionary advantageous characteristics which promote the transformation and progression of phenotypically normal cells into malignant cells over time [34]. The former cancer hallmarks conceived by Hanahan and Weinberg were re-categorized into seven hallmarks, namely, selective growth and proliferative advantage, altered stress response favoring overall survival, vascularization, invasion and metastasis, metabolic rewiring, abetting microenvironment and immune modulation [34].

Hanahan added four new hallmarks in 2022, including two emerging hallmarks and two enabling characteristics, to address the increasing complexity of pathogenesis of cancer over time. They were non-mutational epigenetic reprogramming, unlocking phenotypic plasticity, polymorphic microbiome and senescent cells [35]. The evolution of the cancer hallmarks over time is summarized in Table 1. Cancer cells are capable of undergoing uncontrolled proliferation whereas the hallmarks of cancer are the means that lead to this phenotype. Normal cells can be transformed into cancerous cells by genetic and/or environmental factors. Oncogenic virus with strong epidemiological links to cancers is one of the environmental factors that can be transmitted in utero, perinatally or postnatally. To date, seven oncogenic viruses, namely EBV, human papillomavirus (HPV), hepatitis B virus (HBV), hepatitis C virus (HCV), Kaposi’s sarcoma-associated herpesvirus (KSHV), Merkel cell polyomavirus (MCV or MCPyV) and human T-lymphotropic virus type 1 (HTLV-1), have been identified [36].

Although different oncogenic viruses have their own unique strategies to drive oncogenesis, they have commonalities in hijacking certain cellular pathways. For example, HPV, EBV, HTLV-1, KSHV and MCPyV can modulate phosphoinositol 3-kinases (PI3K)-protein kinase B (AKT)-mammalian target of rapamycin (mTOR) signaling pathway to affect cell growth, proliferation and survival. EBV LMPs facilitate cells in evading apoptosis and acquiring self-sufficiency in growth signals regardless of the availability of nutrients or ligand binding and overcome transforming growth factor beta-1 (TGF-β1)-mediated apoptosis [37,38]. Viral interleukin-6 (vIL-6), a KSHV-encoded cytokine, can unlock phenotypic plasticity of differentiated vascular endothelial cells through the PI3K/AKT pathway. AKT is activated upon vIL-6 binding to the gp130 receptor thereby upregulating prospero homeobox 1 (PROX1) to potentiate lymphatic reprogramming [39]. Oncoprotein E7 of HPV-16 enables cells to invade and metastasize by cytoplasmic retention of cyclin-dependent kinase inhibitor, p27, in a PI3K-AKT-dependent manner [40]. Tax oncoprotein of HTLV-1 activates AKT which phosphorylates forkhead box O3a (FOXO3a) and enables terminally differentiated CD4+ T cells to persist in order to disseminate HTLV-1 [41]. MCPyV small T (sT) antigen promotes hyperphosphorylation of 4E-BP1, a crucial downstream target of mTOR complex 1 (mTORC1), through the PI3K–AKT–mTOR pathway [42]. The above examples demonstrate the contribution of different viral oncoproteins to cancer hallmarks through a single host signaling pathway. An increasing number of studies had revealed the involvement of other viral oncoproteins in the establishment of cancer and defined the corresponding manipulated cellular pathways. Hence, viral-driven oncogenesis is vastly complex and new knowledge remains to be uncovered. A summary of how EBV proteins contribute to the cancer hallmarks is shown in Figure 1. As the functional studies of EBV gene products advance, a conceptual framework is needed to evaluate the contribution of the viral products to oncogenesis systematically.

## 3. Contribution of EBV to Hallmarks of Cancers

### 3.1. Avoiding Immune Destruction

Zta facilitates immune evasion via the downregulation of major histocompatibility complex (MHC) II by transcriptionally repressing immunomodulatory components, class II transactivator (CIITA) and cluster of differentiation (CD) *74* [43,44]. Rta, on the other hand, avoids immune destruction by decreasing interferon (IFN)-β production by suppressing the transcriptional activities of interferon regulatory factors (IRFs) 3 and 7 [45]. BHRF1 dampens innate immunity by inhibiting IFN-β induction through the mitochondrial antiviral signaling protein (MAVS)-stimulator of interferon genes (STING) signaling pathway [46]. Other EBV proteins that have immunomodulatory function are BGLF5, BILF1, BNLF2a, BDLF3, BCRF1, BARF1, BPLF1 and BLRF2. BGLF5 is an exonuclease that is able to modulate immune responses by downregulating the expression of MHC class I and II molecules impairing T-cell recognition as well as reducing toll-like receptor (TLR) 9 levels [47,48,49]. Similarly, BILF1, BNLF2a and BDLF3 disrupt antigen presentation mechanisms impairing CD8+ T-cell recognition of EBV-infected B-cells. BILF1 inhibits both innate and adaptive immune responses by internalizing and depleting MHC I from the cell surface thereby reducing antigen presentation [50,51,52]. BNLF2a blocks the transporter associated with antigen processing (TAP) function, peptide loading and surface expression of MHC class I molecules [53]. BDLF3, a late lytic protein, evades CD8+ and CD4+ T-cell recognition by mediating ubiquitination on MHC molecules [54]. BCRF1 is a viral homolog of human IL-10 (vIL-10) which interferes with the antigen presentation mechanism of MHC class I molecule via downregulation of TAP1 [55,56,57,58,59]. BARF1 is a soluble hexameric glycosylated complex consisting of two immunoglobulin (Ig)-like domains and is detectable in the sera and saliva of NPC patients [60]. It acts as an allosteric decoy receptor that neutralizes and locks human colony-stimulating factor 1 (hCSF1) into an inactive conformation allowing cells to evade immune surveillance [61]. In addition to evading adaptive immune responses via similar mechanisms, BPLF1 and BLRF2 also facilitate evasion of innate immune responses. BPLF1 is a deubiquitylating enzyme (DUB) that suppresses TLR-mediated activation of nuclear factor kappa B (NF-κB) by deubiquitylating IκBα. [62]. BLRF2 inhibits type I IFN production via a cyclic GMP-AMP (cGAMP) synthase (cGAS)-STING pathway [63,64]. It binds to cGAS to inhibit its enzymatic activity and blocks cGAMP synthesis [63]. BPLF1 also deubiquitinates STING and TANK-binding kinase 1 (TBK1) and suppresses cGAS-STING and retinoic acid-inducible gene I (RIG-I)-MAVS pathways [65]. Figure 2 displays the schematic diagram of the role of EBV lytic proteins in modulating immune responses related to oncogenesis.

### 3.2. Activating Tissue Invasion and Metastasis and Inducing or Accessing Vasculature

Activating tissue invasion and metastasis as well as inducing or accessing vasculature are two closely related cancer hallmarks as they are usually modulated by secreted molecules associated with interconnected cellular pathways. Zta directly contributes to angiogenesis, invasion and metastasis via upregulation of cytokines, chemokines and growth factors such as IL-8, vascular endothelial growth factor (VEGF), matrix metalloproteinase (MMP) 3 and MMP9 [66,67,68,69]. Rta modulates the expression of IL-6 and MMP9 whose expression can lead to an increase in tumor invasiveness and metastatic properties [70,71,72]. Other lytic proteins, namely, BILF1, BARF1, BALF1 and BALF3, also contribute to these two cancer hallmarks. BILF1 induces angiogenesis and metastasis through VEGF secretion and intercellular adhesion molecule 1 (ICAM-1) expression [73,74]. BARF1 enhances migration and anchorage-independent growth in human embryonic kidney (HEK)-293 cells [60,75]. BALF1 transfectants exhibited higher rates of haptotactic migration in a transwell migration assay and formed more tumors of larger size in immunodeficient mice [76]. BALF3 expression could enhance metastasis as demonstrated by cell migration, cell invasion and spheroid formation assays [77]. Figure 3 summarizes the roles of EBV lytic proteins in activating tissue invasion and metastasis and inducing or accessing vasculature in cancers along with other hallmarks including genomic instability and mutation, resisting cell death and sustained proliferative signaling.

### 3.3. Genome Instability and Mutation

The cancer hallmark of genomic instability and mutation can be induced by either inhibiting DNA damage response (DDR), repressing DNA repair, inducing DNA damage or interfering with chromosome integrity during cell replication. Early antigen protein D (EA-D), encoded by the BMRF1 gene, is the viral DNA polymerase processivity factor that functions as a transcriptional activator for some EBV and cellular genes [78,79,80]. BMRF1 suppresses DDR by inhibiting the recruitment of RNF168 and the ubiquitylation at double-stranded DNA (dsDNA) breaks [81]. Similarly, BKRF4 inhibits DNA repair and cell signaling associated with dsDNA breaks by binding to histones and blocking the recruitment of RNF168 [82]. In addition to these two proteins, BALF3, BGLF4 and BGLF5 also induce genetic alterations by promoting the formation of micronuclei and chromosomal abnormality, inducing DNA damage or DNA strand breaks and repressing the repair of DNA damage [77,83,84]. Moreover, Rta can induce genomic instability in epithelial cells by causing chromosome mis-segregation through the activation of extracellular signal-regulated kinases (ERK) signaling [85]. BNRF1 is another lytic protein that contributes to chromosomal aberrations by mediating the degradation of the structural maintenance of chromosomes (SMC) protein 5/6 (SMC5/6) [86,87]. Overexpression of BNRF1 can lead to aneuploidy [86]. In addition, it can also disrupt the formation of the DAXX-ATRX chromatin remodeling complex which potentially supports B-cell transformation [88].

### 3.4. Resisting Cell Death, Sustaining Proliferative Signaling and Other Cancer Hallmarks

A major characteristic of cancer lies in its ability to survive and proliferate owing to two related cancer hallmarks, namely, resisting cell death and sustaining proliferative signaling. BARF1 can activate B-cell lymphoma 2 (BCL-2) expression which leads to the elevation in the ratio of BCL-2 to BCL-2-associated X protein (BAX) and reduces Poly(ADP-Ribose) Polymerase 1 (PARP1) cleavage to protect the cells from apoptosis [89,90]. Other EBV proteins that have anti-apoptotic properties include BHRF1 and BALF1 which are Bcl-2 homologs [91,92,93]. Inhibition of pro-apoptotic proteins by BHRF1 was shown to facilitate chemoresistance and protect cells from apoptotic stimuli upon treatment with apoptosis-inducing agents [94]. BHRF1 also contributes to the cancer hallmark of unlocking phenotypic plasticity as epithelial cell differentiation is perturbed upon ectopic expression of BHRF1 in a human squamous cell carcinoma line [91]. On the other hand, BALF1′s role in inhibiting apoptosis may vary depending on the virus life cycle [92,95]. In addition, Zta can bind to the promoter of tumor necrosis factor (TNF) receptor 1 (TNFR1) and downregulate its expression to prevent TNF-α-induced apoptosis [96]. Our group previously showed that Zta might modulate autophagy initiation [97] which is a common mechanism that enables cell survival and drug resistance in cancers. Moreover, Zta downregulates the expression level of more than 2000 cellular genes during EBV lytic replication including those genes responsible for immune response, apoptosis signaling and lymphocyte activation [98]. In contrast, genes responsible for sustaining proliferative signaling, such as those encoding for paracrine and autocrine growth factors, are upregulated [98]. BLLF3, an EBV early gene encoding deoxyuridine triphosphate nucleotidohydrolase (dUTPase), facilitates tumor-promoting inflammation by inducing the expression of the miRNA-155 [99]. Virally induced miRNA-155 expression was shown to be critical for the growth of EBV-positive lymphoblastoid cell lines (LCLs) during latency [100]. In addition, BLLF3 induces inflammation by utilizing a TLR2-dependent mechanism to stimulate the release of pro-inflammatory cytokines such as IL-1β, TNF-α and IFN-ɣ [101,102,103,104].

## 4. Postulated Oncogenic Roles of EBV Lytic Proteins

The roles and function of a large number of EBV lytic proteins in the viral life cycle and oncogenesis have remained uncharacterized. One approach to assess the possible function of EBV lytic proteins is to examine the function of their homologs in closely related viruses or other oncoviruses. For example, BNRF1 interacts with DAXX at the PML nuclear body (PML-NB)/nuclear domain 10 (ND10) to disrupt the formation of the DAXX-ATRX complex [88]. Although no interaction between open reading frame (ORF) 75 of KSHV (homolog of BNRF1) and DAXX was demonstrated, depletion of ATRX and dispersion of DAXX were observed [105]. These findings indicate that different herpesviruses employed distinct mechanisms in counteracting the repression caused by ND10 [105]. BNRF1 also targets the intrinsic viral DNA immune sensor, SMC5/6, for proteasomal degradation to evade its suppression and facilitate the formation of lytic replication compartments [87]. Similar to BNRF1, KSHV RTA and Rta homologs of other gamma-herpesviruses could also target the SMC5/6 complex for degradation [106]. In addition, HBV HBx protein promotes the degradation of SMC5/6 by hijacking the cellular DNA damage-binding protein 1 (DDB1)-containing E3 ubiquitin ligase [107,108,109]. E2 protein of HPV also interacts with the SMC5/6 complex which inhibits the viral replication initiation function of E2 [110,111]. Since the viral proteins of diverse oncoviruses may have evolutionarily conserved and shared functions, examining the functions of homologs amongst different oncoviruses may provide an effective strategy to assess the potential oncogenic mechanisms of EBV lytic proteins. Figure 4A shows the similarity of functions between BNRF1, KSHV RTA, HBV HBx and HPV E2 in modulating cellular mechanisms that promote genomic instability and mutation.

An additional function of BMRF1 in the degradation of PARP1 through a proteosome-dependent mechanism was identified by aligning to the function of other viral processivity factors (PF) such as KSHV PF-8 and MHV68 mPF [112,113]. Since PARP1 is a repressive host protein that negatively regulates lytic replication in gamma-herpesviruses, degradation of PARP1 is a viral strategy to facilitate lytic replication and inhibit DDR [114,115]. On the other hand, KSHV PF-8 could enhance DNA replication through its interaction with protein arginine methyl transferase 5 (PRMT5) which leads to an open chromatin formation [116]. PRMT5 is considered to play a role in cancer progression through the modulation of various oncogenic cellular pathways such as cell proliferation, migration and invasion [117]. KSHV PF-8 interacts with KSHV long noncoding polyadenylated nuclear RNA (PAN RNA) to recruit host demethylases, namely, ubiquitously transcribed tetratricopeptide repeat on chromosome X (UTX) and Y (UTY) [118,119,120]. HPV E7 can also upregulate UTX and UTY causing virally-induced epigenetic reprogramming [121,122]. In addition to its role in inducing genomic instability via DDR inhibition, BMRF1 also interacts with the nucleosome remodeling and deacetylase (NuRD) complex which may result in the transcriptional activation of oncogenes [81]. The MTA3 subunit of the NuRD complex interacts with BCL-6, an oncogene that has transformative potential in diffuse large B-cell lymphoma (DLBCL) [123,124]. Similarly, the methyl-CpG-binding domain protein 3 (MBD3) subunit can interact with transcription factor JUN, an oncogene that plays critical roles in several cancers [125]. The oncogenic mechanisms of BMRF1, KSHV PF-8 and HPV E7 are summarized in Figure 4B. Whether BMRF1 contributes to cancer development through the manipulation of the oncogenic properties of NuRD or through a similar strategy as that of KSHV awaits future investigations.

EBV SM protein is a product of spliced BSLF2 and BMLF1 open reading frames that functions in mRNA transport. It is essential in the production of infectious viral particles particularly in supporting the expression of gp350/220 and other late lytic proteins [126,127]. Homologs of SM in other herpesviruses include infected cell protein 27 (ICP27) of herpes simplex virus (HSV), unique long 69 (UL69) of human cytomegalovirus (HCMV) and ORF57 of KSHV and herpesvirus saimiri (HVS) [128,129,130,131]. These viral homologs maintain RNA stability, modulate RNA splicing, control protein expression and, in some instances, act as RNA export factor [132]. HVS ORF57 was able to substitute SM in trans-activating gene expression in a transient expression assay containing the DR enhancer region of EBV showing the functional similarity of these two proteins [133]. Similarly, SM could compensate the viral replication defects of ICP27-null HSV-1 virus [128]. Interestingly, KSHV ORF57 seemed to be functionally distinct from the other homologs in its RNA splicing role as it functions as a viral splicing factor whereas HSV ICP27 is a splicing inhibitor [134,135,136,137]. While the role of SM as a cellular export factor has been established [138,139,140,141], its role in viral and cellular RNA splicing varies [142,143,144,145,146,147]. Nonetheless, several pathogenic roles have been suggested based on the posttranscriptional regulatory function of SM homologs in herpesviruses. KSHV ORF57 enhances the expression of a VEGF receptor, kinase insert domain receptor fetal liver kinase-1 (KDR/flk-1) which can increase cell proliferation and vasculature [148]. HSV ICP27 induces disruption of transcription termination of RNA polymerase II (RNAPII) via interactions with an mRNA 3′ processing factor known as cleavage and polyadenylation specificity factor (CPSF) [149]. SM, however, inhibits cell proliferation and induces the expression of IFN-stimulated genes [150]. SM induces the expression of signal transducer and activator of transcription 1 (STAT1) and several interferon-stimulated genes (ISGs) in EBV-negative B cells and epithelial cells in which STAT1 has been linked to the induction of cellular senescence [151,152,153]. This puts forth the idea that cellular senescence can become a hallmark of EBV carcinogenesis. Figure 5 shows a schematic diagram of EBV SM protein’s known and postulated contribution to cancer hallmarks with the latter inferred from the function of its homologs in KSHV and HSV and from that of other oncoviral proteins.

In addition to its roles in post-transcriptional regulation, SM mediates global SUMOylation (small ubiquitin-like modifier, SUMO) of host proteins [154]. SM, HCMV UL69 and HSV ICP27 can all upregulate global SUMOylation via either SUMO1 or SUMO2 supporting the importance of SUMOylation in herpesvirus infection [154]. They all have E3 SUMO ligase activity and can SUMOylate p53 in vitro and in vivo [154]. HPV E6/E7 oncoproteins could also manipulate the SUMOylation pathway by impairing autophagy resulting in the accumulation of ubiquitin conjugating protein (Ubc9) which, in turn, leads to evasion of apoptosis in premalignant cells [155]. HPV E6 lowers Ubc9 levels to decrease global SUMOylation [156] whilst E7 modulates the SUMOylation of an oncogenic transcription factor, Forkhead box M1b (FoxM1b) by interacting with Ubc9 and protein inhibitor of activated STAT 1 (PIAS1) [157]. The dysregulation of FoxM1b function leads to disruption in cell cycle and genomic stability [157]. Adenovirus E1B-55K and E4-ORF3 proteins induce SUMOylation of p53 and transcription intermediary factor 1γ (TIF-1γ), respectively [158,159,160]. Similarly, KSHV also SUMOylates p53, Rb and its own basic region-leucine zipper (bZIP) protein through its SUMO2/3-specific E3 ligase activity [161]. Growing evidence suggests that SUMOylation may play an important role in cancer owing to its function in modulating gene expression, maintaining genome integrity and regulating cell cycle progression [162]. Considering the diversity of cancer hallmarks being modulated by SM and its homologs, the validation of their oncogenic roles and development of therapeutic targeting of these proteins in virus-associated cancers will be warranted.

## 5. Conclusions

This review has summarized the known oncogenic properties of EBV lytic proteins in the context of four subcategories of cancer hallmarks to which they contribute, namely, (1) evading immune responses, (2) tissue invasion and metastasis and inducing or accessing vasculature, (3) genome instability and mutation and (4) resisting cell death and sustaining proliferative signaling. New oncogenic roles of lytic proteins of EBV are postulated by examining their homologs for shared function in other herpesviruses or oncoviruses. The evolutionarily conserved viral proteins among different viruses of the same family can signify their importance in providing survival or growth advantages for virus replication or replication of virally-infected cells. In addition, a shared cellular function that promotes oncogenesis may be identified amongst different virus families. This article provides a conceptual framework of inferring oncogenic properties of EBV proteins by cross-comparing with those of functional homologs in other herpesviruses and oncoviruses. This framework may provide the basis to drive functional and oncogenic investigations of oncoviral proteins.

While looking into homologs of viral proteins may provide insight in the potential roles of EBV lytic proteins in oncogenesis, some challenges will be encountered particularly if homologs of closely-related viruses have apparently contradictory functions. Having the framework of cancer hallmarks to map the oncogenic potential of viral proteins can serve to categorize their pathogenic roles, guide the direction of research on novel pathways associated with the new cancer hallmarks and potentially translate the findings to novel therapeutics against EBV- and other virus-associated cancers.

## Figures and Tables

**Figure 1 cancers-15-02120-f001:**
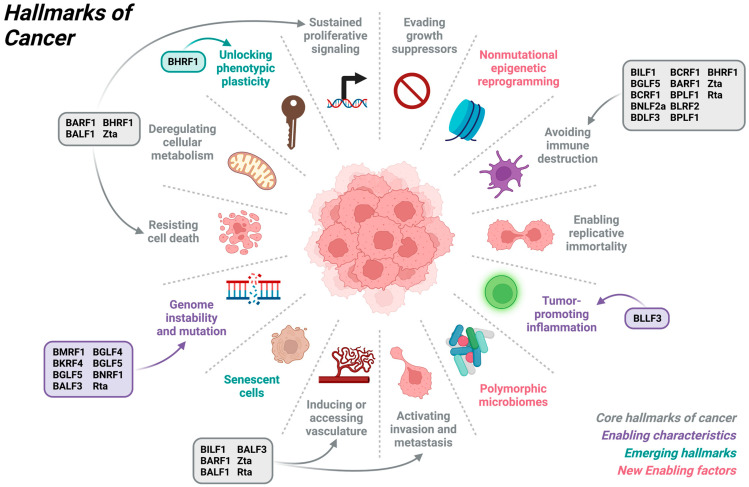
Contribution of EBV lytic proteins to hallmarks of cancers. EBV lytic proteins directly contribute to the hallmarks of avoiding immune destruction, activating invasion and metastasis, inducing or accessing vasculature, genome instability and mutation, resisting cell death, unlocking phenotypic plasticity and sustained proliferative signaling. Created with BioRender.com. Figure was adapted from “Hallmarks of Cancer: Circle” by BioRender.com (2023). Retrieved from https://app.biorender.com/biorender-templates, accessed on 23 February 2023.

**Figure 2 cancers-15-02120-f002:**
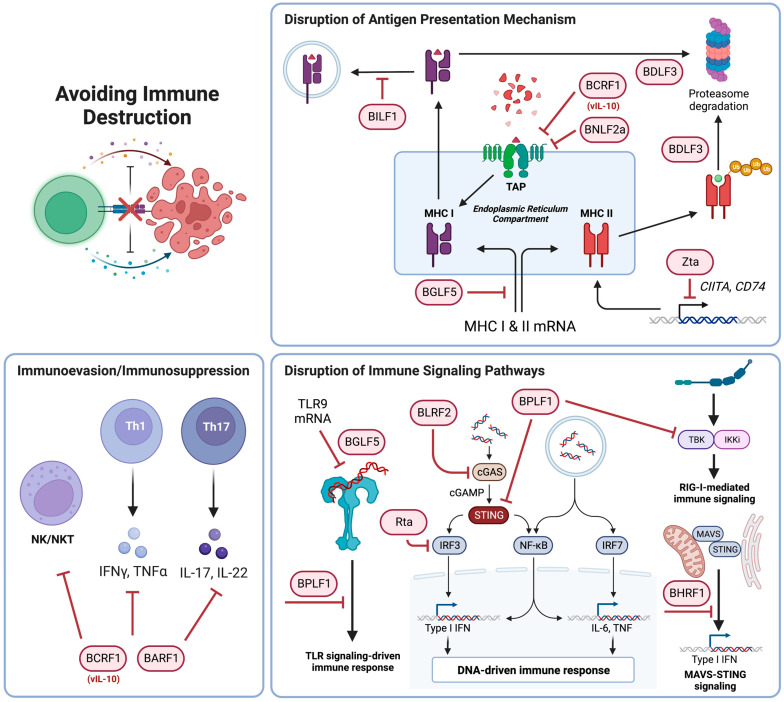
Schematic diagram of the mechanisms employed by EBV to evade immune responses. Zta, BGLF5, BILF1, BNLF2a, BDLF3 and BCRF1 avoid immune surveillance by modulating the antigen presentation mechanisms. BCRF1 and BARF1 are inhibitory molecules that facilitate immune evasion. Rta, BHRF1, BGLF5, BLRF2 and BPLF1 evade innate immune responses by modulating various components of immune signaling pathways. Created with BioRender.com. Figure was adapted from “Icon Pack—Cytokine” and “cGAS-STING DNA Detection” by BioRender.com (2023). Retrieved from https://app.biorender.com/biorender-templates, accessed on 24 February 2023.

**Figure 3 cancers-15-02120-f003:**
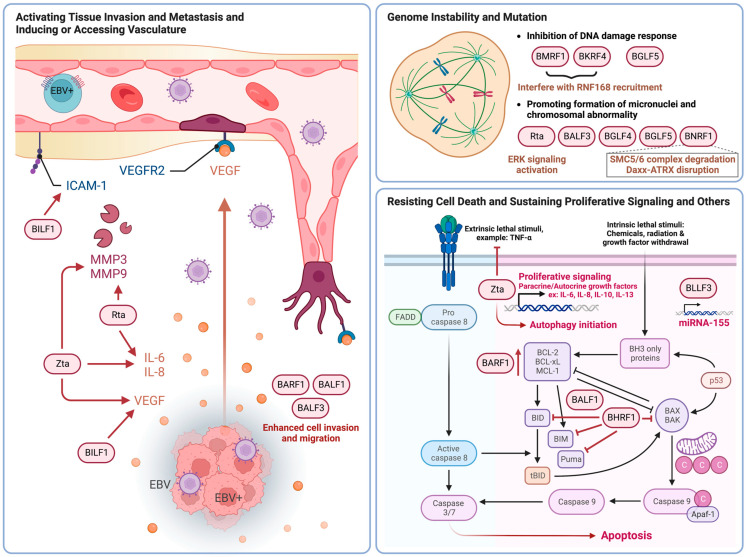
Schematic diagram of strategies of EBV in activating tissue invasion and metastasis, inducing or accessing vasculature, promoting genomic instability and mutation, resisting cell death and sustaining proliferative signaling. Zta, Rta, BILF1, BARF1, BALF1 and BALF3 induce tissue invasion, metastasis and angiogenesis. BMRF1, BKRF4, BALF3, BGLF4, BGLF5 and BNRF1 induce genomic instability and mutation through the inhibition of DNA damage response (DDR) and the formation of micronuclei and chromosomal instability. Zta, BARF1, BALF1 and BHRF1 promote cell survival via either cell death resistance or sustained proliferative signaling. Created with BioRender.com. Figure was adapted from “Tumor Vascularization” and “Extrinsic and Intrinsic Apoptosis” by BioRender.com (2023). Retrieved from https://app.biorender.com/biorender-templates, accessed on 24 February 2023.

**Figure 4 cancers-15-02120-f004:**
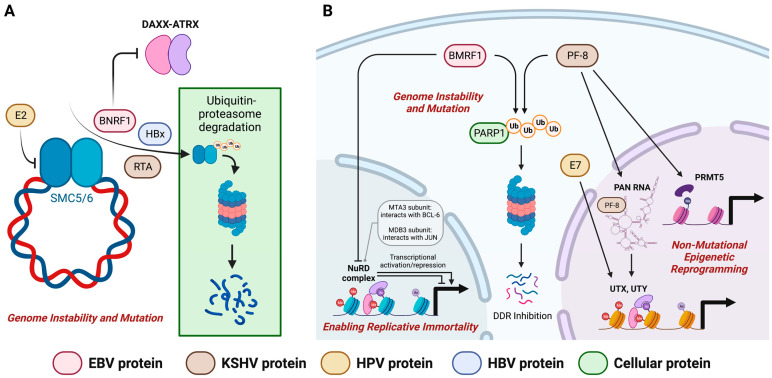
Molecular mechanisms of BNRF1 and BMRF1 and their homologs of other oncoviruses in contributing to hallmarks of cancer. (**A**) BNRF1, HBV HBx, KSHV RTA and HPV E2 modulate DNA damage response (DDR) in cancer cells to promote genomic instability and mutation. (**B**) BMRF1 and KSHV PF-8 degrade PARP1 to inhibit DDR. Interaction between BMRF1 and NuRD may regulate both transcriptional activation and repression of cellular proteins enabling replicative immortality in cancer cells. KSHV PF-8 and HPV E7 can modulate the expression of PRMT5, UTX and UTY to promote epigenetic reprogramming. Created with BioRender.com.

**Figure 5 cancers-15-02120-f005:**
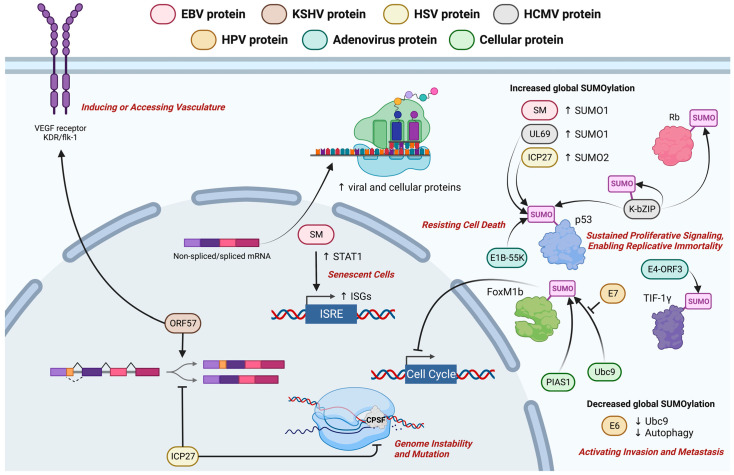
Schematic diagram of EBV SM protein’s known and postulated contribution to cancer hallmarks. SM increases SUMOylation and STAT1 expression though its role in EBV oncogenesis is unclear. Several postulated hallmarks of cancer according to the function of SM homologs are inducing or accessing vasculature, resisting cell death, sustained proliferative signaling, enabling replicative immortality, senescent cells, genome instability and mutation and activating invasion and metastasis. Created with BioRender.com.

**Table 1 cancers-15-02120-t001:** Hallmarks of cancer proposed by Hanahan and Weinberg in 2000, 2011 and 2022.

2000 [32]	2011 [33]	2022 [35]
Evading apoptosis	Deregulating cellular energetics	Non-mutational epigenetic reprogramming
Self-sufficiency in growth signals	Avoiding immune destruction	Unlocking phenotypic plasticity
Insensitivity to anti-growth signals	Genomic instability and mutation	Polymorphic microbiome
Sustained angiogenesis	Tumor-promoting inflammation	Senescent cells
Limitless replicative potential		
Tissue invasion and metastasis

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
