# Peer review of "Contribution of Epstein–Barr Virus Lytic Proteins to Cancer Hallmarks and Implications from Other Oncoviruses"

_cancers, 2023, doi:10.3390/cancers15072120_

Round 1

Reviewer 1 Report

This review focuses on the relationship between “Cancer Hallmarks”, a well-known carcinogenic process, and “Epstein-Barr virus (EBV)”. It is quite unique concept as EBV review paper. Especially, this review’s figures are excellent compared to previous EBV reviews in this journal.

However, to be accepted as a review article, the quality of the English text needs to be significantly improved.

1.     Abbreviation

There are numerous abbreviation errors in this review. All these must be improved. For example, Epstein-Barr virus is described twice in Lane 33 and Lane 111. Similar issues abound in this review. Conversely, there are also parts where abbreviations appear without full spellings such as EBNA-1 and BART in Lane 48.

2.     Comma

In general expression, it is written as “A, B, and C”. However, in this paper, there is also a description of "A, B and C".

3.     Long sentences

The most important problem is that there are many long sentences.

Lane 51-55

These distinctive expression patterns confer survival advantages for EBV since EBV-transformed cells in latency express fewer EBV gene products at their cell surface, allowing them to evade cytotoxic immune responses.

Lane 59-61

While EBV latent proteins have been demonstrated to drive cellular transformation and tumorigenesis, increasing evidence suggests that EBV lytic proteins are equally important in contributing to EBV-driven oncogenesis.

It is very difficult to understand such a long sentence accurately in a short time. Therefore, it is strongly recommended to use the simpler representation. And also use high quality English editor, who has a knowledge for biochemistry or virology.

Reviewer 2 Report

In this excellent and comprehensive review of lytic proteins of EBV and cancer, the authors have done a tremendous job of summarizing the various proteins involved in many key "Hallmarks of Cancer." This provides a very nice framework along with summarizing key homologies across oncogenic viruses. They have summarized the important literature and put into context quite effectively. I have no recommendations for specific edits or changes and recommend acceptance.

Reviewer 3 Report

This is a well written and presented manuscript. However, there are several mirror comments that in my opinion need to be addressed prior to acceptance.

On lines 23 and 41, the authors indicate that EBV exhibits “biphasic life cycle: lytic and latent” which is the classical assumption. However, there is increasing evidence that a third phase probably exists in vivo; an abortive lytic phase in which progeny virus is not produced. While the authors indicate such  phase has been reported (lines 68-70) only a single reference (22) is provided. In my opinion the potential role of abortive lytic replication in oncogenesis should be expanded and a more extensive list of references included (J Virol 86:7976; 2012; Cancers 10:98; 2018; Adv Exp Med Biol 1225:127-135, 2020; Microorganisms 8:1824; 2020).

Figure 1: BLLF3 should be included as tumor promoting inflammation. A recent review concerning the potential role of BLLF3 in carcinogenesis has been published (Cancers 15:855; 2023) and this should be indicated in the references.

Line 230 “Early antigen protein D (EA-D) encoded by the BMRF1 gene….The authors use the term EA-D throughout the manuscript (lines 300, 302, 303, 309, 329, 335, 336) and interchangeable with BMRF-1 (Fig 1 and 3). The term EA-D originally referred to Early Diffuse Antigen (EA-D) which is actually a complex of proteins (J Virol 47:193, 1983; J Virol 53:793; 1985; J Med Virol 79:1710, 2007). Except for reference 105 in the manuscript which defines “EA-D as being encoded by the BMRF-1 gene” the remaining references (76,77,78,116) use BMRF-1. The authors should remove the use of EA-D and replace with BMRF-1 though out the manuscript.

Round 2

Reviewer 1 Report

It has no problem in resubmitted version.